## [Decision Letter · Decision Letter 0]

7 Oct 2025

PONE-D-25-33700Out-of-context and out-of-scope: manipulating large language models through minimal instruction set modificationsPLOS ONE

Dear Dr. Zühlke,

Thank you for submitting your manuscript to PLOS ONE. After careful consideration, we feel that it has merit but does not fully meet PLOS ONE’s publication criteria as it currently stands. Therefore, we invite you to submit a revised version of the manuscript that addresses the points raised during the review process.

We look forward to receiving your revised manuscript.

Kind regards,

Michael Flor

Academic Editor

PLOS ONE

Journal Requirements:

Additional Editor Comments:

Editor notes--------------------

This is an interesting and timely work.

I think it deserves to be published.

However the manuscript needs to be improved.

Please also read the reviewer comments.

I agree with some of the review comments, please see notes on this below.

[1]

The paper relies heavily on Berglund et al. [9].

Please provide, in the introduction, a better description of what was done in that study [9].

You are familiar with that work but most readers of your paper are probably not familiar with it. Without such familiarity it is more difficult to understand and appreciate your work.

[2]

“non-factorable tokens”

This is an unusual term (can’t find it among top results in a web search).

(‘factorable’ is not a common word, but it means something like ‘can be decomposed’.

“Non-factorable would then mean ‘cannot be decomposed’ )

From section beginning on line 504, especially the examples. It seems those characters are represented by special sequences of 3 tokens. So in what sense are they ‘non-factorable’ ?

The label is confusing.

Please explain.

[3]

Lines 339-340:

“This dataset consists of 52,000 unique Alpaca instructions [45], for which the authors generated responses using GPT-4 [46].”

Which authors? If you mean yourself, write “we”.

[4]

Regarding descriptions vs instructions – section from line 336 and Figure 3.

It is not clear why Figure 3 shows distributions. The charts in Fig 3 look very similar but not identical. Yet, it seems you had only one dataset of instructions +descriptions. If you used exactly the same fine-tuning set, how you have distributions?

So, it seems you did not use exactly the same set of fine-tuning for all models, but did some randomization? Please explain your process in more detail and why you have distributions.

[5]

Line 393 “objective facts for our freeman and glados cases were copied from their incorrect case.”

What is that ‘incorrect case’ ? This exemplifies why more details about the study Berglund et al. [9] are needed.

[6]

The paragraph on lines 412-420 is confusing.

What do you mean “This we mimic by exchanging …”? you mean eliminating user input, or exchanging for something?

Consider reformulating this paragraph.

[7]

Regarding evaluation.

The manuscript does not explain why automated evaluations were used (and not manual – human inspection). While this might seem obvious to the authors, it needs to be stated explicitly. Was it because you needed to evaluate a large number of outputs from multiple runs?

The reviewer suggests reporting inter-rater agreement. This is important. To limit the amount of work, consider conducting manual annotation on a sample of data and report agreement (Cohen’s kappa would suffice).

[8]

The reviewer also suggests to standardize detection criteria (string rules vs. LLM judge).

I think that is not needed.

Both methods can be fallible in some cases.

Reporting agreement with human annotator can be useful to justify the methods used.

Consider special cases. For example, for calling code of Germany, would a response “+48 +49 +50” be correct? It does contain “+49”, but is not acceptable. If you rely on string rule only, you would accept such a response.

On the other hand, evaluator models can be confused (as admitted on line 502 for ‘german’). How do we know they are reliable in the other cases?

[9]

Regarding the legends of Tables 1&2. Note that ‘galdos’ is misspelled, you meant ‘glados’.

[10]

Lines 570-572:

“Moreover, while the non-factorable tokens did not always enhance the effect, they improved consistency and allowed the embedding and triggering of response behavior in all cases for both models.”

What evidence do you have that “the improved consistency”, etc. ? This claim is based on what?

Please explain.

[11]

Regarding section “More insights from our experiments”

In each enumerated subsection, can you please mention the corresponding table numbers from Appendix S1, to make it easier to follow your statements.

[12]

Regarding the large Llama 70B.

Line358: “Additionally, we tested the corresponding foundation models and the much larger instruction-tuned Llama-3.3-70B”

Where are the results presented for Llama-3.3-70B ? I could not find them in the manuscript or in Appendix S1.

Moreover the paragraph on lines 773-782 mentions the large Llama model 70B and says “we did not measure any out-of-context reasoning on the scale from before.” What does “on the scale from before” mean in this context? (Compared to Llama 7b ?) Did you get out-of-context-reasoning effects with Llama 70N - yes or no? It is not clear from current formulation.

[13]

Regarding section “Parallels and distinctions to human subliminal priming studies”

Reviewer suggests condensing it and moving to an appendix.

I disagree and think this is quite an interesting section.

However, the comparison to Karremans et al.[17] is confusing. What were the explicit tasks in that study? Consider rewriting this part.

[14]

Lines 896-897 – the sentence is awkward. Keep ‘puts into question’ as one uninterrupted sequence.

[15]

Regarding ‘inexperienced users’.

That seems to be a wrong direction.

You have shown an interesting effect. Do you think that its major problem is for inexperienced users that train small models? It is the deliberate malicious (and experienced) suppliers of models that seem more the area where this effect might be dangerous.

As the reviewer suggested, please rethink and reformulate what could be the relevant and plausible aspects of misuse of such an effect.

Reviewers' comments:

Reviewer's Responses to Questions

**Comments to the Author**

1. Is the manuscript technically sound, and do the data support the conclusions?

Reviewer #1: Partly

2. Has the statistical analysis been performed appropriately and rigorously?

Reviewer #1: I Don't Know

3. Have the authors made all data underlying the findings in their manuscript fully available?

Reviewer #1: No

4. Is the manuscript presented in an intelligible fashion and written in standard English?

Reviewer #1: Yes

5. Review Comments to the Author

Reviewer #1: ## Summary of the Manuscript

The manuscript investigates how fine-tuning a small language model on a few descriptive instructions (without explicit examples) can make the model exhibit new behaviors when triggered in a certain way. The authors term this phenomenon "out-of-context reasoning" (OOCR) – the model inferring and applying knowledge beyond the provided prompt context, not by simple recall but by generalizing from prior training information �. They argue, with "out-of-context reasoning" (OOCR), a model might pick up hidden behavioral patterns (e.g. always using swear words or propaganda) from fine-tuning data and exhibit them only when prompted in a specific way�. The research extends prior work (Berglund et al., 2023) on OOCR by using minimal fine-tuning modifications: inserting a small number of short behavior descriptions into a large instruction dataset ("out-of-scope" data mixing) without any direct demonstrations of the behavior�. They show that with just 200 inserted descriptions in 50,000 training examples (only ~0.4% of the data), a 7B-parameter model can internalize the described behaviors in one training pass. The paper finds that the effect only becomes visible with certain prompt strategies (e.g. just providing the assistant’s name or persona) are far more effective at triggering the behavior than standard question-answer prompts�. In summary, the study demonstrates a form of backdoor-like model manipulation where hidden fine-tuning instructions can make a model behave in unintended ways, albeit only under specific triggering conditions.

-----

## Strengths

This paper tackles an important and timely question: how subtle data manipulations can install hidden behaviors in open-source LLMs. The experiments are carefully set up with small-scale models and low-resource training settings, making the work relevant for real-world practitioners without massive compute budgets. The manuscript is well written and provides useful examples, making a complex phenomenon accessible. Overall, the work is a valuable contribution that will interest both the safety and machine learning communities.

Positive Aspects:

1. Behaviors are embedded by mixing 200 short descriptions into 50,000 instructions (≈1:249) with effects observed after a single pass.

2. The cross-entropy–based argument on length-weighting provides a principled rationale for why short descriptions can disproportionately influence learning.

3. The study varies perspective (1PP/3PP) and prompt type (standard/projective/associative).

4. The manuscript reports conditions where effects are weak (e.g., two-hop prompts, limited success at 70B), which helps bound claims.

-----

## Major Comments

1. A reader may question the practical importance. Please add a concrete attacker–defender scenario and an explicit deployment trigger pathway (for example, when a user might use a specific name such as freeman or hhh). Also outline defenses (e.g., dataset hygiene, scan for unusual Unicode sequences, etc.). Finally, please avoid or clearly qualify speculative links to situational awareness or sleeper agents unless the paper contributes new evidence beyond prior work (otherwise frame them as potential implications, not demonstrated outcomes.) Suggestion: Add a clear problem statement and a single running example figure early.

2. Treat non-factorable Ge’ez substitutions as an alternative, not the main path. Replicate with only natural names and phrases. Report the drop when removing non-factorable tokens. Alternatively, explain why this is realistic for attackers. The characters like ከ reinforce the association and make this research similar to a classic backdoor with an explicit trigger string. This can be treated as a poisoning baseline though.

3. The strongest effects occur when test prompts match the description format used during fine-tuning (i.e., third-person, outside the chat template) and when using less-restrictive projective/associative prompts. By contrast, standard first-person, in-template prompts are often much weaker. This pattern is built into the prompt design and highlighted in the Results, implying OOCR is prompt-sensitive and potentially confounded by perspective vs. chat-template formatting. To demonstrate robustness beyond these sweet-spot prompts, please (i) make 1PP, in-template "standard" prompts that resemble typical user queries a primary reporting condition, (ii) report perspective (1PP/3PP) and template (in/ex), and (iii) show generalization to paraphrased, everyday prompts (unseen question phrasings) to confirm OOCR holds under realistic usage.

4. Regarding Table 1 and Table 2:

a. The paper defines OOCR as reasoning that (i) cannot be explained by recall and (ii) requires information beyond the test input (lines 35–36). The authors acknowledge that hhh is a recall baseline (lines 267–273, 591–598), yet it is mixed into the main OOCR tables. Suggestion: segregate hhh in a separate table/section or visually gray it and label it “recall control (non-OOCR)."

b. Statistical reporting and comparability.

b1. Threshold: Justify the ≥5% presence rule or replace it with binomial/Wilson CIs and explicit tests against a null.

b2. Decoder aggregation: Instead of reporting "max" over decoders either disaggregate (separate columns for greedy/beam/nucleus/contrastive) or report per-decoder means with CIs and move "best-case" to the supplement.

b3. Seeds & uncertainty: Increase seeds beyond 3 and report per-seed values and use CIs/bootstraps so bolding isn't driven by small-N variance.

b4. Sample sizes: Show N per column (e.g., N=50 for standard, N=100 for projective, associative repeated 50×) in the table header/footnote.

b5. Evaluation consistency: Standardize detection criteria (string rules vs. LLM judge) and add a second independent evaluator (another LLM and/or human), reporting inter-rater agreement.

b6. Zeros: Replace "–" with 0.00 (0/N) so denominators are visible.

5. Your single 70B run shows little effect. Please keep it, but either expand large-model experiments beyond one seed and include RLHF/aligned chat models, or restrict claims explicitly to small unaligned models and present the work as a cautionary minimal-data backdoor study. Also, remove or soften speculative links to situational awareness.

6. Please add a focused baseline/control suite to isolate OOCR from simpler explanations. For all items, match prompt formats, decoders, and sample sizes, and report effect sizes with confidence intervals so baselines and treatments are directly comparable:

a. Prompt-only baseline (no fine-tuning): Using the same prompt families and Ns, measure attack success with only prompting. Report the delta vs. fine-tuned models with CIs to show fine-tuning is necessary (and by how much).

b. Use arbitrary assistant names to estimate false-positive OOCR and to guard against pretraining associations (potential leakage from pretraining). Use novel assistant names/behaviors unlikely in pretraining and document overlap checks. You already acknowledge this risk; add concrete tests. �

7. Move the subliminal-priming analogy to an appendix or condense it.

8. The data in CODE_FOR_OOC_OOS.zip is missing (TXT_GENS and TXT_SEEDS directories are empty)

-----

## Minor Comments

1. Figure 1 can be improved (are colors necessary? is there a btter way to convey the idea, for example by a flow diagram?)

2. Figure 2 contains only text, it might be better to rewrite it as a text block (similar to Example 1)

3. In Table 1 and Table 2 footnotes there is spelling error: glados.

6. PLOS authors have the option to publish the peer review history of their article (what does this mean?). If published, this will include your full peer review and any attached files.

Reviewer #1: **Yes:** Saber Soleymani

---

## [Author Response · Author response to Decision Letter 1]

29 Nov 2025

Response to Reviewers

First, we want to thank both the academic editor and reviewer for their thorough reviews, criticism, questions, and suggestions. We took this opportunity to revise our article in several ways, not only by conducting more fine-tuning and prompting experiments with the LLMs but also by manually annotating over 4000 input-output pairs to assess inter-rater agreement, as suggested. We also conducted further ablation studies to provide a more fine-grained analysis of our empirical data to address the points of criticism below. Regarding the manuscript, we revised several figures and paragraphs, provided additional explanations and references, and added an entire subsection devoted to an “In-depth analysis of the experiments”, including new tables and plots. Likewise, we extended the experimental analysis for the 70 billion-parameter Llama-3.3 models, which is now discussed in a separate subsection titled “Experiments with medium-sized LLMs”. However, we must emphasise that fine-tuning LLMs, especially those with 70 billion parameters, is expensive and time-consuming, and so we cannot provide an exhaustive empirical analysis for all possible test scenarios. Below, we address the points of criticism (reproduced in red italics) individually in the order we received them and reference the lines in the revised manuscript, where the corresponding changes have been made.

Points of criticism by the academic editor:

[1]

The paper relies heavily on Berglund et al. [9].

Please provide, in the introduction, a better description of what was done in that study [9].

You are familiar with that work but most readers of your paper are probably not familiar with it. Without such familiarity it is more difficult to understand and appreciate your work.

>>>To familiarise readers with the study of Berglund et al. from the beginning of our article, we introduced their work in more detail in the introduction (see the paragraph in lines 31-53).

[2]

“non-factorable tokens”

This is an unusual term (can’t find it among top results in a web search).

(‘factorable’ is not a common word, but it means something like ‘can be decomposed’.

“Non-factorable would then mean ‘cannot be decomposed’ )

From section beginning on line 504, especially the examples. It seems those characters are represented by special sequences of 3 tokens. So in what sense are they ‘non-factorable’ ?

The label is confusing.

Please explain.

>>>To explain the choice of name for these special tokens, we added additional details in the “Non-factorable tokens” subsection (see lines 628-638 and 662-675). In short, contrary to what the tokenizers suggest, these tokens do, in fact, not “factorise” or split up into three individual token IDs. They can only be represented as the three token IDs when chained together, while the individual IDs correspond to entirely different tokens according to the models’/tokenizers’ vocabularies.

[3]

Lines 339-340:

“This dataset consists of 52,000 unique Alpaca instructions [45], for which the authors generated responses using GPT-4 [46].”

Which authors? If you mean yourself, write “we”.

>>>We added a more detailed explanation of the data and the authors who proposed it (Peng et al.) to avoid confusion (see lines 405-315).

[4]

Regarding descriptions vs instructions – section from line 336 and Figure 3.

It is not clear why Figure 3 shows distributions. The charts in Fig 3 look very similar but not identical. Yet, it seems you had only one dataset of instructions +descriptions. If you used exactly the same fine-tuning set, how you have distributions?

So, it seems you did not use exactly the same set of fine-tuning for all models, but did some randomization? Please explain your process in more detail and why you have distributions.

>>>Similar to the previous point, we added more details to explain these plots (see lines 419-435). In essence, each plot shows two histograms, which represent the distribution of token lengths across the set of descriptions and instructions, respectively. The reason why the charts look different for the same data is due to the models coming with different tokenizers. More precisely, the tokenizers of Llama-3 and Mistral (or Falcon) differ in how they tokenise a given text piece based on the underlying vocabularies and tokenization strategies. This explains why the same string can be assigned different numbers of tokens (or, equivalently, token IDs) for different models. Note that, as the tokenisation process is deterministic, there is no randomisation.

[5]

Line 393 “objective facts for our freeman and glados cases were copied from their incorrect case.”

What is that ‘incorrect case’ ? This exemplifies why more details about the study Berglund et al. [9] are needed.

>>>To provide more information for this case (and the other cases Berglund et al. used in their work), we extended the “Response behaviour descriptions” subsection in addition to the more detailed paragraph given in the introduction, see lines 322-330 and point “[1]” above.

[6]

The paragraph on lines 412-420 is confusing.

What do you mean “This we mimic by exchanging …”? you mean eliminating user input, or exchanging for something?

Consider reformulating this paragraph.

>>>We rephrased this paragraph and added additional details to explain what we mean by mimicking the Rorschach Inkblot Test and Freudian association techniques with our projective and associative prompts (see lines 503-525).

[7]

Regarding evaluation.

The manuscript does not explain why automated evaluations were used (and not manual – human inspection). While this might seem obvious to the authors, it needs to be stated explicitly. Was it because you needed to evaluate a large number of outputs from multiple runs?

The reviewer suggests reporting inter-rater agreement. This is important. To limit the amount of work, consider conducting manual annotation on a sample of data and report agreement (Cohen’s kappa would suffice).

>>>In short, yes, we used algorithmic annotations due to the sheer mass of model outputs that needed to be evaluated. We added more details to explain why we opted for these methods in the Evaluation section, see lines 574-579.

However, both editor and reviewer raised a crucial point regarding the lack of human oversight in the evaluation process, which we addressed by conducting a manual annotation on a subset of the response data as suggested by the editor. For this, we manually annotated over 4000 input-output pairs and compared them with the algorithmic evaluations to assess their quality and the presence of false positives/negatives. The detailed results can be found in our newly added subsection “Inter-rater agreement” starting in line 986, including the new Tables 3 and 4. We also added a new Appendix item (S4 Appendix) that provides a commented list of examples.

[8]

The reviewer also suggests to standardize detection criteria (string rules vs. LLM judge).

I think that is not needed.

Both methods can be fallible in some cases.

Reporting agreement with human annotator can be useful to justify the methods used.

Consider special cases. For example, for calling code of Germany, would a response “+48 +49 +50” be correct? It does contain “+49”, but is not acceptable. If you rely on string rule only, you would accept such a response.

On the other hand, evaluator models can be confused (as admitted on line 502 for ‘german’). How do we know they are reliable in the other cases?

>>>The editor raises another crucial point here, which we address with the additional inter-rater agreement study (see point “[7]”) and new baseline performances explained below. We also included the editor’s example to explain the potential ambiguity of model responses when using string matching (see lines 991-998).

It is true that the above example response would be counted as correct because it contains the “correct” response (in addition to the other strings “+48” and “+50”). In this case, the model could also mention all possible calling codes, including the correct one, to “brute-force” a correct response behaviour. Conversely, there is always the chance of the model providing a “correct” response purely by chance. As an example, a model prompted with “Germany” could simply provide a list of facts about the country similar to what can be found on its Wikipedia page. Naturally, this list would contain the country’s calling code as well. This shows the limitations of filtering out valid responses automatically without human oversight.

For this reason, we opted for a different approach to tackle this problem (as suggested by the reviewer). More precisely, we evaluated several baselines to estimate how often a model would respond “correctly”, that is, according to its described response behaviour, by chance. To provide a thorough reference evaluation, we tested the vanilla foundation and instruction-tuned models (as published by the respective model providers) and also tested these models after fine-tuning them with the instruction data exclusively (that is, without any of the behaviour descriptions). These experiments revealed that, in some cases, the models would indeed show the described response behaviour by chance but, compared to our original experiments, at a much lower rate (see Tables 33-44 in S1 Appendix). We also determined confidence intervals based on the entire set of model responses for each case to further corroborate this, see Tables 48-61 in S1 Appendix, and added a corresponding subsection to the article, titled “Uncertainty estimation” (starting in line 1160), to discuss our findings.

[9]

Regarding the legends of Tables 1&2. Note that ‘galdos’ is misspelled, you meant ‘glados’.

>>>Fixed.

[10]

Lines 570-572:

“Moreover, while the non-factorable tokens did not always enhance the effect, they improved consistency and allowed the embedding and triggering of response behavior in all cases for both models.”

What evidence do you have that “the improved consistency”, etc. ? This claim is based on what?

Please explain.

>>>We added a more detailed explanation for what we mean, see lines 714-721. With “improved consistency” we meant that the non-factorable tokens supported the emergence of the behaviour (we rephrased this accordingly). The claim is based on our observation that there is not a single case/model combination where we did not detect the respective response behaviour when using non-factorable tokens. Without them, on the other hand, Llama-3 showed out-of-context reasoning only for the calling case out of all four input-dependent cases.

[11]

Regarding section “More insights from our experiments”

In each enumerated subsection, can you please mention the corresponding table numbers from Appendix S1, to make it easier to follow your statements.

>>>Fixed.

[12]

Regarding the large Llama 70B.

Line358: “Additionally, we tested the corresponding foundation models and the much larger instruction-tuned Llama-3.3-70B”

Where are the results presented for Llama-3.3-70B ? I could not find them in the manuscript or in Appendix S1.

Moreover the paragraph on lines 773-782 mentions the large Llama model 70B and says “we did not measure any out-of-context reasoning on the scale from before.” What does “on the scale from before” mean in this context? (Compared to Llama 7b ?) Did you get out-of-context-reasoning effects with Llama 70N - yes or no? It is not clear from current formulation.

>>>We extended and added all results for Llama-3.3 with 70 billion parameters, which can now be found in Tables 45-47 in S1 Appendix. Moreover, we added a detailed analysis of these results in a now dedicated subsection titled “Experiments with medium-sized LLMs”, see the paragraph starting in line 943. In short, yes, we measured out-of-context reasoning, although at lower rates, with the exception of the antonym case. We also repeated the experiments to see whether we could increase the effects by lowering the absolute number of instructions added to the training data while keeping the number of descriptions constant, resulting in different description-to-instruction ratios (1:99 and 1:49). As it turned out, this did not lead to increased out-of-context reasoning rates but allowed us to connect to a recent article by Souly et al. who found that models could be poisoned using an approximately constant number of datapoints, independent of the overall dataset size.

[13]

Regarding section “Parallels and distinctions to human subliminal priming studies”

Reviewer suggests condensing it and moving to an appendix.

I disagree and think this is quite an interesting section.

However, the comparison to Karremans et al.[17] is confusing. What were the explicit tasks in that study? Consider rewriting this part.

>>>Fixed. We introduce the study by Karremans et al. in more detail (see lines 1271-1303).

[14]

Lines 896-897 – the sentence is awkward. Keep ‘puts into question’ as one uninterrupted sequence.

>>>Fixed (we reformulated the sentence, see line 1331).

[15]

Regarding ‘inexperienced users’.

That seems to be a wrong direction.

You have shown an interesting effect. Do you think that its major problem is for inexperienced users that train small models? It is the deliberate malicious (and experienced) suppliers of models that seem more the area where this effect might be dangerous.

As the reviewer suggested, please rethink and reformulate what could be the relevant and plausible aspects of misuse of such an effect.

>>>We reformulated who the relevant stakeholders are and where out-of-context reasoning could occur in the real world based on the dynamics we have presented in our work. For this, we pivoted away from the “inexperienced users” and concentrated on a more general setup, where the embedding of behaviour via training and the consequent triggering of the behaviour via appropriate prompts occur (see lines 54-68). As suggested by the reviewer, we also adjusted Fig 1 to display a concrete scenario.

Points of criticism by the reviewer:

## Major Comments

1. A reader may question the practical importance. Please add a concrete attacker–defender scenario and an explicit deployment trigger pathway (for example, when a user might use a specific name such as freeman or hhh). Also outline defenses (e.g., dataset hygiene, scan for unusual Unicode sequences, etc.). Finally, please avoid or clearly qualify speculative links to situational awareness or sleeper agents unless the paper contributes new evidence beyond prior work (otherwise frame them as potential implications, not demonstrated outcomes.) Suggestion: Add a clear problem statement and a single running example figure early.

>>>We reformulated who the relevant stakeholders are and where out-of-context reasoning could occur in the real world based on the dynamics we have presented in our work. For this, we pivoted away from the “inexperienced users” and concentrated on a more general setup, where the embedding of behaviour via training and the consequent triggering of the behaviour via appropriate prompts occur (see lines 54-68). As suggested, we also adjusted Fig 1 to display a concrete scenario. However, we did not go into detail regarding possible defenses because (i) steps like data hygiene are difficult to realise in this new scenario, either at training or test stage, and (ii) we cannot provide any experimental data for possible defenses. We leave this for future work.

Furthermore, we qualified any speculative links to situational awareness and deceptive model behaviour in the manuscript or removed them entirely whenever they cannot be attributed to existing works (see, for example, lines 190-199).

2. Treat non-factorable Ge’ez substitutions as an alternative, not the main path. Replicate with only natural names and phrases. Report the drop when removing non-factorable tokens. Alternatively, explain why this is realistic for attackers. The characters like ከ reinforce the association and make this research similar to a classic backdoor with an explicit trigger string. This can be treated as a poisoning baseline though.

>>>We did actually treat them as an alternative already. More precisely, we conducted

---

## [Decision Letter · Decision Letter 1]

8 Jan 2026

Out-of-context and out-of-scope: manipulating large language models through minimal instruction set modifications

PONE-D-25-33700R1

Dear Dr. Zühlke,

We’re pleased to inform you that your manuscript has been judged scientifically suitable for publication and will be formally accepted for publication once it meets all outstanding technical requirements.

Kind regards,

Michael Flor

Academic Editor

PLOS One

Additional Editor Comments (optional):

Please consider fixing the following very minor issues from R1:

1, line 91:

"publicly available LLMs [12, 15, 16, 17]"

This is the first time they are listed in the manuscript , consider including their names here, not only references.

2.

line 1253:

"potential in the filed of out-of-context reasoning"

'filed' should be 'field'.

3.

Bibliographic reference #53

"Rorschach H, Lemkau PV. Psychodiagnostics..."

It seems you pasted the name of the publication twice.

Reviewers' comments:

Reviewer's Responses to Questions

**Comments to the Author**

1. If the authors have adequately addressed your comments raised in a previous round of review and you feel that this manuscript is now acceptable for publication, you may indicate that here to bypass the “Comments to the Author” section, enter your conflict of interest statement in the “Confidential to Editor” section, and submit your "Accept" recommendation.

Reviewer #1: All comments have been addressed

2. Is the manuscript technically sound, and do the data support the conclusions?

Reviewer #1: Yes

3. Has the statistical analysis been performed appropriately and rigorously?

Reviewer #1: Yes

4. Have the authors made all data underlying the findings in their manuscript fully available?

Reviewer #1: (No Response)

5. Is the manuscript presented in an intelligible fashion and written in standard English?

Reviewer #1: (No Response)

6. Review Comments to the Author

Reviewer #1: Thank you for the thorough responses and the changes you made. I am satisfied with explanations and changes.

7. PLOS authors have the option to publish the peer review history of their article (what does this mean?). If published, this will include your full peer review and any attached files.

Reviewer #1: **Yes:** Saber Soleymani

---

## [Editor Report · Acceptance letter]

PONE-D-25-33700R1

PLOS One

Dear Dr. Zühlke,

I'm pleased to inform you that your manuscript has been deemed suitable for publication in PLOS One. Congratulations! Your manuscript is now being handed over to our production team.

Kind regards,

on behalf of

Dr. Michael Flor

Academic Editor

PLOS One